# Drift Algal Accumulation in Ice Scour Pits Provides an Underestimated Ecological Subsidy in a Novel Antarctic Soft-Sediment Habitat

**DOI:** 10.3390/biology12010128

**Published:** 2023-01-13

**Authors:** Ignacio Garrido, Heather L. Hawk, Paulina Bruning, Luis Miguel Pardo, Ladd E. Johnson

**Affiliations:** 1Department of Biology and Québec-Océan, Laval University, Québec, QC G1V 0A6, Canada; 2Centro FONDAP de Investigación Dinámica de Ecosistemas Marinos de Altas Latitudes (IDEAL), Valdivia 5090000, Chile; 3Laboratorio Costero de Recursos Acuáticos de Calfuco (LCRAC), Instituto Ciencias Marinas y Limnológicas (ICML), Facultad de Ciencias, Universidad Austral de Chile, Valdivia 5110566, Chile

**Keywords:** Antarctic Peninsula, Antarctic benthic communities, polar benthos, polar warming, drift algae, ice pits, ecological subsidy, ice scouring

## Abstract

**Simple Summary:**

Whereas several studies have documented the destructive effects of ice scouring on the seabed and its strong role in structuring benthic communities in Antarctica, none have highlighted the potential of seabed furrows (i.e., “ice pits”) produced by this disturbance as a novel habitat that is created by the accumulation of drift algae. This work documents the distribution and dimensions of ice pits to evaluate their relationships with the biomass and species composition of the accumulations of drift algae within them. Moreover, the temporal dynamics of algal deposition and advective loss in ice pits over time are assessed. These investigations show that ice pits provide hitherto underestimated ecological subsidies of food and shelter for other benthic organisms. Further research is needed to better understand the role of ice pits in the overall functioning of Antarctic marine benthic ecosystems.

**Abstract:**

Ice scouring is one of the strongest agents of disturbance in nearshore environments at high latitudes. In depths, less than 20 m, grounding icebergs reshape the soft-sediment seabed by gouging furrows called ice pits. Large amounts of drift algae (up to 5.6 kg/m^2^) that would otherwise be transported to deeper water accumulate inside these features, representing an underestimated subsidy. Our work documents the distribution and dimensions of ice pits in Fildes Bay, Antarctica, and evaluates their relationship to the biomass and species composition of algae found within them. It also assesses the rates of deposition and advective loss of algae in the pits. The 17 ice pits found in the study area covered only 4.2% of the seabed but contained 98% of drift algal biomass, i.e., 60 times the density (kg/m^2^) of the surrounding seabed. Larger ice pits had larger and denser algal accumulations than small pits and had different species compositions. The accumulations were stable over time: experimentally cleared pits regained initial biomass levels after one year, and advective loss was less than 15% annually. Further research is needed to understand the impacts of ice scouring and subsequent algal retention on ecosystem functioning in this rapidly changing polar environment.

## 1. Introduction

The disturbance is one of the most important factors that regulate the structure and functioning of natural communities [1,2,3,4,5]. In marine communities, several physical disturbance processes have been explored for intertidal [6,7,8,9,10,11] and subtidal systems [12,13,14,15] in temperate latitudes. At the high latitudes of Antarctica, marine environments are exposed to these natural physical disturbances (e.g., storm events [16]; sedimentation [17,18]) but also to those associated with the movement of ice [19,20,21] and glacial melting [22,23]. Their combined effects strongly shape benthic ecosystems [24]. 

Ice, a dominant disturbance agent at high latitudes, is found primarily in two forms: sea ice and icebergs [25]. Sea ice is formed when the surface of the water freezes and often covers large areas as either pack ice floating at the surface or as landfast ice attached to the shore. During the seasonal break-up, sea ice can scour the shoreline to depths of several meters. In contrast, icebergs are larger floating masses of ice that have detached from glaciers or ice shelves. The relevant difference between the two is that icebergs are much thicker, and thus they can disturb the bottom at much deeper locations, even to the depth of 100s of meters [26,27,28]. Both forms of ice can drift by wind, currents, and tides and have direct mechanical, thermal, physical, and chemical impacts on the Antarctic coastal bottom [29].

Disturbance from icebergs can shape seabed morphology [30], reworking both hard and soft benthic seabeds [31]. In particular, contact of icebergs with the soft bottom produces deep furrows in the seabed [32,33,34,35,36]. Known as “ice plow marks” [37], “ice scores” [38], “ice scour” [39,40], “ice scour tracks” [41], and “ice gouge” [42,43], the terminology for these features has, unfortunately, not been well standardized [26]. Here we use the term “ice scouring” to refer to the disturbance process of either iceberg being dragged across the seabed or sea ice pushed against the shoreline by wind, waves, or currents. We distinguish “ice pits” as the depressions left on soft bottoms by this process. The formation of ice pits coincides with the seasonal formation and movements of icebergs. Ice scouring occurs as pulsed events with varying frequency and intensity, but in general, icebergs are formed in winter and then, after being pushed by winds, ocean currents, tidal changes, and storms, they become grounded on the coastal seafloor during early spring [44]. Various studies have qualitatively described the modification of the otherwise featureless soft sediment environments by this disturbance, producing heterogeneous benthic topographies [34,35], but none have finely mapped the distribution, shapes, and sizes of such features. 

Several studies have clearly documented the destructive effects of ice scouring on the benthic communities in Antarctica on both hard and soft bottoms [45]. Examples include a >90% reduction in species abundance and >60% reduction in species richness around Adelaide Island [46] and an even greater reduction (>99%) in the macrofaunal abundance in a soft sediment habitat at Signy Island [47]. However, no studies have highlighted the potential of this form of disturbance to create a novel habitat on soft bottoms by creating depositional environments for unattached material. In particular, macroalgae detached from the substratum or fragmented by physical disturbance agents such as wave action [48,49] and biological factors such as grazing or reproduction [50] can accumulate on the bottom as drift algae [49,51,52], including in polar waters [35]. In subpolar regions, the production of drift algae is similar to that of temperate shores [53,54], being driven primarily by wave action, especially during storms. However, in higher latitudes (i.e., polar regions), this process can also be dominated by ice scouring [46,55,56].

Once algae are detached, their transport is driven mainly by currents (tidal- and wind-generated), but depending on their structural components and buoyancy, some dislodged algae may float and drift at the ocean surface or water column for long periods of time before accumulating on the shore [57] or sinking to the bottom [58,59]. Material that sinks will generally be carried by bottom currents to locations where bottom features prevent further displacement [35] and allow the deposition and accumulation of drifting material. Once accumulated on the bottom, the fate of drift algae is unknown but will depend on abiotic and biotic conditions [60]. If there is sufficient light for gas exchange, photosynthesis, growth, and even reproduction are possible [61,62]; however, degradation is more likely. Algae may be consumed by fauna, and it decomposes in place, especially in thicker depositions, where microbial respiration forms hypoxic or anoxic conditions in bottom layers [63]. Finally, the advective loss is also possible if hydrodynamic conditions change, e.g., an increase in current or storm-generated surge.

The accumulation of macroalgae in ice pits could have multiple implications for the functioning of coastal ecosystems [35,64]. First, it could represent an important coastal carbon sink for primary production that would otherwise be carried offshore [65]. Second, it could create a habitat for local fauna such as amphipods, isopods, and fish [66,67]. Third, the algae could provide a food resource for herbivores and detritivores, which support higher trophic levels [68,69]. Indeed, ice pits have the potential to form a distinct module of the coastal food web, retaining locally produced primary production and creating physical structure. Finally, ice pits are local sinks for the storage of blue carbon and represent potential locations for carbon sequestration via the burial of algal material. 

Given these potential ecological roles of ice pits and their potentially widespread distribution around Antarctica, further knowledge of their physical and biological characteristics is needed to assess their importance, especially as an ecological subsidy for benthic soft-bottom communities [70]. This information is crucial to improve our understanding of the processes that control the patterns of distribution and diversity at high latitudes [21,22,71]. Our current lack of information is not surprising due to the challenges of underwater research, especially with the inaccessibility and logistical difficulty of working in the Southern Ocean [72,73]. However, given the predicted changes in response to global warming [74], it is urgent to describe and document this unique habitat both to understand its potential ecological roles as a food resource, habitat, and carbon storage and to provide a baseline for future changes [75] in the formation and distribution of ice pits. The dynamics of ice-generated disturbance are predicted to change dramatically with warming temperatures that will affect both the formation of sea ice and the production of icebergs [76].

Here we provide a detailed survey of a subtidal site in the West Antarctic Peninsula where we conducted descriptive sampling and experiments to document the distribution and nature of ice pits and drift algae, including species composition and biomass. Specifically, we (1) documented how ice pits are distributed in space, (2) described their physical dimensions, (3) evaluated biomass and species composition of drift algae within ice pits, and (4) determined the temporal dynamics of algal deposition and accumulation in ice pits through time.

## 2. Materials and Methods

### 2.1. Study Site

The study was conducted in Fildes Bay at a site near the Instituto Antártico Chileno (INACh) Prof. Julio Escudero research station on King George Island, Antarctica (Figure 1). The site (62°12′16.8″ S, 58°56′58.6″ W) was located in the 300-m-wide channel formed between the coast and Albatros Islet (Figure 2). The maximum depth in the channel is 20 m but increases to the east into the bay. The subtidal bottom on both sides of the channel is characterized by a moderate slope of medium-sized boulders to a depth of 3 m, after which the slope, consisting primarily of bedrock and boulders, steepens until a depth of 10 m. At greater depths, the substratum changes to sandy sediment with occasional pebbles. Attached algae occur primarily in the upper 10 m. In the first 3 m of depth, small patches of *Adenocystis utricularis* (Ochrophyta), *Monostroma hariotii* (Chlorophyta), and *Palmaria decipiens* (Rhodophyta) occur. At greater depths, abundant patches of algae attached to rocks appear, mainly the rhodophytes *Plocamium cartilagineum*, *Curdiaea racovitzae,* and *Palmaria decipiens*. The phaeophytes *A. utricularis* and *Himanthorallus grandifolius* and the chlorophyte *M. hariotii* are also present but are less abundant. On the deeper soft bottom, attached algae are not found, except occasionally attached to the shells of the infaunal bivalves *Laternula elliptica* and *Aequiyoldia eightsii* or small rocks. In this zone, accumulations of drift algae comprise the overwhelming majority of algal material.

Fildes Bay regularly freezes in winter; therefore, field aspects of this study, which spanned 2018–2020, were conducted during the austral summer (January and February) when the bay was ice-free and allowed access for untethered SCUBA diving.

### 2.2. Distribution and Characterization of Ice Pits

The distribution of ice pits at the study site was mapped in 2018 using 15 transects that were run perpendicular to the coastline of the islet. Transects were 50 m long and 5 m wide, covering in total an area of 3750 m^2^ that ranged in depth from 12 to 20 m. The center of each pit encountered was used to measure the bathymetric depth and the depth of the ice pit (Figure 3), and a buoy was floated to the surface for georeferencing (GARMIN GPSmap 78s; Garmin Ltd., Olathe, KS, USA; Table 1). The length and width of each pit were measured along the centermost axes, except for ice pits longer than 2 m, where additional width measurements were taken to calculate a mean width (Figure 3). Because the exact demarcation of the edge of the ice pits was often difficult to judge, we used the more distinct edge of the accumulated algal layer within the ice pit for our measurements of length and width. As these measurements were smaller than the physical dimensions of the ice pits, our estimates for ice pit dimensions are thus slightly underestimated (see dashed line in Figure 3A). The thickness of the algal layer was also measured at the center of the pit. From these dimensions, the algal accumulation area (modeled as an ellipse) and algal accumulation volume (modeled as a hemi-ellipsoid) were estimated (Figure 3).

### 2.3. Drift Algae Composition among Ice Pits

Due to time constraints, eleven of the 17 surveyed ice pits were randomly chosen to describe the composition of drift algae that accumulated within ice pits. Within each one, two 0.25-m^2^ quadrats were haphazardly selected. For each quadrat, we collected all the algae material and some underlying sediment using an airlift device (suction dredge sampling) constructed with an auxiliary SCUBA cylinder connected through a BCD hose and an air feed valve to the wall of a 2-m-long, 10-cm-diameter polyvinylchloride (PVC) pipe manipulated by SCUBA divers (Appendix A). When the air was released from the cylinder, it sucked material into one end of the pipe, which was then retained within a bag (2-mm mesh) attached to the other end of the pipe (see also [23,71]). Two quadrats were also collected from haphazardly chosen locations within 4 m of the edge of each ice pit (Figure 3C).

Once transferred to the surface, samples were stored in separate mesh bags in coolers before being transported to the nearby INACh laboratory, where the contents of each bag were deposited separately in aquaria supplied with circulating seawater at a constant temperature of 1 °C. Within 24 h, algae were then identified and sorted to the lowest taxonomic level possible [77,78]. Samples of the separated species were then spun for 20 s in a kitchen salad spinner to remove excess water [79] and then weighed to the nearest 0.01 g. The mean biomass of the two quadrats were used to calculate the area-specific biomass (ASB; kg/m^2^), which was multiplied by the thickness of the algal layer to calculate the algal density (AD; kg/m^3^). The total algal biomass for each ice pit was then estimated by multiplying algal density by the calculated algal accumulation volume (Table 1). 

### 2.4. Drift Algae Temporal Dynamics

The temporal dynamics of drift algae in ice pits were assessed by two experimental manipulations. In order to estimate rates of algal deposition, drift algae were removed annually in January from five ice pits over three years (2018–2020). The cleared ice pits were randomly selected from among the smaller (≤10 m^2^) ice pits in the study area for logistic reasons (e.g., the volume and transport of removed material). By using the suction technique described above, all the algae in each pit were removed and weighed in January 2018, i.e., at the beginning of the summer sampling season. At the start of the two subsequent summers (January 2019 and 2020), the algal accumulation in each ice pit was again removed and weighed. In addition, during the intervening summers (2018 and 2019), the pits were visited two, four, and six weeks after the major clearing event, and any accumulated algae was removed and weighed. 

In order to estimate rates of advective loss of algae from ice pits, ten markers were haphazardly placed on top of the algal layer in seven ice pits and then censused after one month and again after 13 months. The ice pits were randomly selected from among those not used in the clearing experiment. The markers were made of strips of orange-colored straight-line polyvinyl chloride (PVC) flagging tape, each 15 cm long and 2 cm wide. As well as being highly visible for recovery (Figure 3D), the markers had a high surface-volume ratio and near-neutral buoyancy (density = 1.95 g/cm^3^), which made them suitable proxies for fragments of drift algae.

### 2.5. Statistical Analysis

Pearson’s correlations were used to assess relationships among ice pit dimensions (length, width, and depth). A Mantel test was used to determine if physical proximity between ice pits (i.e., Euclidean distance in meters, along the orthogonal axes of the sampling grid, and bathymetric depth) contributed to similarities in ice pit shape (i.e., Euclidean distance in meters along axes of ice pit length, width, and depth). A Welch *t*-test was used to distinguish the area-specific biomass inside ice pits from that of the surrounding seabed.

Biotic metrics, including total algal biomass (kg), the area-specific algal biomass (kg/m^2^), the algal species richness (S), and diversity (Shannon-Wiener’s H′), were calculated and compared among ice pits. To determine if ice pit dimensions and relative position were good predictors of these metrics, they were modeled using multiple regressions. Environmental variables with relevant interaction terms include bathymetric depth (m), ice pit depth (m), and local coordinates (m). For the latter, latitude and longitude were converted to a meter-based grid within the study area so that all spatial measurements and coefficients were interpreted in meters; locations corresponded to distances east and north of the reference point based on the most southwesterly ice pit “L” and “M”). Initial models for the area-specific biomass, species richness, and H′ diversity also included ice pit length (m), mean width (m), and additional interaction terms. The final models of the four responses were determined by first removing multicollinearity (threshold of VIF = 4.6), then reducing predictors and interaction terms based on a stepwise threshold of ΔAIC of 0.1, and finally, by inspecting model residuals.

To determine if the similarity in ice pits dimensions corresponded to the similarity in relative algal species composition, a Mantel test was performed between the Euclidean distance matrix of ice pit shape (i.e., length, width, and depth) and the Euclidean distance matrix based on the biomass of each of the 16 algal species. The relationship was also visualized using non-metric Multidimensional Scaling (nMDS) based on Bray-Curtis dissimilarity, which included algal species composition correlated with physical variables (environmental fitting).

A one-way analysis of variance (ANOVA) was performed to test whether annual algal accumulations measured in 2019 and 2020, each year after being experimentally cleared, differed from the biomass initially observed in the 2018 survey. Annual accumulation values were summed from all removal events, including year-long and short-term (two-, four-, and six-week) collections. The Bartlett test was used to verify the homogeneity of variances for biomass as a function of observation year, and the Shapiro-Wilks test was used to verify the normality of model residuals. Rates of advective loss of PVC markers were calculated after one and 13 months but were not statistically compared. Analyses were performed using R software (version 4.0.2; R Development Core Team, Vienna, Austria, 2020), and *p* < 0.05 was used as the significance threshold for all tests.

## 3. Results

### 3.1. Distribution and Characterization of Ice Pits

In the surveyed area, 17 ice pits were found and geo-referenced. They were, however, not randomly distributed across the surveyed area: all were found at depths between 15 and 19 m, with most falling in two north-south running bands (a western one of 10 ice pits and an eastern one of five ice pits) that ran diagonally to the shorelines (Figure 2). Ice pits varied greatly in all physical dimensions with ranges almost double the mean values (mean [range]: length = 5.1 m [1.0–10.1]; width = 2.0 m [0.8–4.3]; and depth = 1.9 m [0.5–3.5]). Their dimensions were, however, not strongly correlated: length and width were moderately correlated (r = 0.59, *p* = 0.01), whereas ice pit depth was not related to either length (r = 0.14, *p* = 0.6) or width (r = 0.36, *p* = 0.16). There was almost a 30-fold range in the calculated value for the ice pit area (mean [range]: 9.27 m^2^ [0.79–21.8]). The Mantel test indicated that spatial autocorrelation did not contribute to the variation in ice pit shape and size (Table 2), i.e., similarities in ice pit dimensions were not related to physical proximity.

### 3.2. Drift Algal Biomass and Composition with Respect to Ice Pit Shape

Algal biomass was almost two orders of magnitude greater (*t* = 7.29, *p* < 0.001) inside the ice pits (mean ± sd = 2.92 kg/m^2^ ± 0.19) than outside ice pits (0.05 kg/m^2^ ± 0.05). Sixteen algal species were found in the samples (Table 3), but three species, *Plocamium cartilagineum*, *Curdiaea racovitzae* (both rhodophytes), and *Himantothallus grandifolius* (a large, endemic phaeophyte), contributed 81.5% of the total biomass (Figure 4; Table 3). These species were also the most commonly observed drift algae outside of the ice pits.

Among ice pits, there was substantial variation in biomass, both in terms of total algal biomass and area-specific biomass. Total algal biomass varied over two orders of magnitude (0.4–58.7 kg) among the 11 ice pits sampled. Not surprisingly, ice pits that were longer and, therefore, larger in the area had greater total algal biomass (Table 4), but they also had greater area-specific biomass (*p* < 0.001 and R^2^ = 0.89 and R^2^ = 0.71 for length and area, respectively (Figure 5), which varied over an order of magnitude among the ice pits. Analysis by multiple regressions showed a significant interaction between the east-west spatial location and bathymetric depth (Table 4) for total algal biomass and a significant effect of ice pit length on area-specific biomass. Because of the low variation in the thickness of the algal layer (all but two were 15 cm thick; Table 1), algal densities closely followed the pattern of area-specific biomass.

In terms of drift algal diversity, ice pit size (specifically length) and bathymetric depth both significantly affected species richness, with higher species richness in larger ice pits located in deeper water (Table 4). No factor was found to influence the Shannon-Wiener diversity index (analysis not shown). The algal species composition among ice pits was also found to correspond closely to the similarity in ice pit dimensions (Mantel r = 0.57, *p* = 0.001) (Table 2). These results were visualized with an nMDS plot where most larger ice pits (L, Q, S)) were separated by length and width and dominated by two of the most abundant drift algal species, *Plocamium cartilagineum* and *Himantothallus grandifolius* (Figure 6). In contrast, most smaller ice pits (B, I, E) were well separated from other ice pits, especially along the first dimension of the plot, with their species composition dominated by rarer species, specifically *Georgiella confluens*, *Desmarestia anceps*, and *Ascoseira mirabilis*. Moderate-sized ice pits (A, C, G) also formed a distinct group, separated from the larger ice pits by their geographic location (eastern band of ice pits) and the higher relative abundance of another common species, *Curdiea racovitzae*.

### 3.3. Drift Algae Temporal Dynamics

Initial biomass values of accumulated drift algae were almost completely restored one year after each experimental removal, with no statistical differences seen among the three years (F = 0.0889, df = 2, *p* = 0.92; mean biomass of 2018, 2019, and 2020 = 9.71 kg, 9.2 kg, and 9.36 kg, respectively; Figure 7). Almost none of this recovery occurred during the summer removals: on average, ice pits regained only 68 g after two weeks and an additional 45 g after four weeks; no additional algae were observed after six weeks. Advective loss of drift algae was estimated to be very low. Only two of the PVC markers were lost after 45 days (loss rate of 2%/month), and more than 85% were recovered in the next year (loss rate of 1%/month).

## 4. Discussion

In Antarctica, ice pits of various sizes are created in the soft sediment of the seabed by icebergs, disturbing infaunal communities yet creating a unique habitat following the accumulation of drift algae in the ice pits (this study) and the subsequent colonization by associated invertebrates (I. Garrido, unpublished data). Our 17 study ice pits were found in a small study area (approximately half the area of a football, aka soccer, field), covering just over 4.2% of the seabed but containing over an estimated 98% of the drift algal biomass. How representative this situation is of other nearshore environments remains unknown, but ice pits are also present at other sites in Fildes Bay (e.g., Nelson and Collins glaciers; Figure 1).

Whether similar degrees of drift algal accumulation happen in ice pits that are in shallower or deeper water than the Fildes Bay study area has not been measured. However, previous studies in Antarctica described iceberg scouring on soft bottoms at depths down to 500 m [27,31]. Moreover, scouring by sea ice and very small icebergs can occur at depths shallower than those in this study, where source populations of algae are more likely to be found due to light limitations [80]. Thus, the rates of the dislodgment of algae by ice scour, their subsequent transport, and their accumulation on the seabed remain unknown but are key variables in this equation. Given the vast areas of the Antarctic continental shelf, which averages 500 m in depth and is largely soft sediment [81], the role of ice pits in the physical and biological processes of benthic ecosystems may be greatly underestimated.

The frequency and intensity of ice-scouring events are determined by the regional production of icebergs [82,83], the oceanographic forces that regulate their movements (i.e., tides, winds, and currents), and local geomorphology. The frequency of ice pit formation in Fildes Bay has not been previously described, and no newly created ice pits were observed during the study. Likewise, no existing ice pits appeared to diminish or disappear, even pits of only 1 m^2^ in area. Admittedly, the three years of the study may be a short time frame for examining geological processes, but the observations suggest that ice pits are formed rarely, or perhaps episodically, and remain long-lived features. This study, albeit limited in spatial and temporal extent, should serve as an impetus for further work, both along routes of detailed, small-scale studies as well as systematic larger-scale seafloor mapping that take advantage of modern technologies such as side-scan sonar [84].

Studies such as this can also serve as a baseline for documenting the impacts of future changes to this environment, in particular those associated with climate change [85]. Over decadal timescales accelerated rises in temperatures will increase the production of icebergs as glaciers, and ice shelves melt more rapidly [83]. The fate of more numerous icebergs is unpredictable, but likely outcomes include increases in grounding and ice scouring with subsequent increases in both algal dislodgment and ice pit formation. On the other hand, higher rates of disturbance may favor small, ephemeral green algae that do not contribute large amounts of drifting biomass—indeed, very few species in this study were ephemeral, suggesting that the current scales of disturbance allow living algal assemblages to reach late successional status with large and long-lived individuals. Over centennial timescales, the disappearance of ice shelves and the retreat of tidewater glaciers will lead to fewer icebergs and, thus, fewer ice pits, and eventually, this unique habitat could simply disappear.

This study also highlights the physical diversity of ice pits, in particular, their geometric dimensions. Both length and width were highly variable but correlated, suggesting that wider icebergs are dragged farther across the seabed. Surprisingly, neither dimension was correlated with pit depth, perhaps reflecting a limit to the extent that icebergs can penetrate into the seabed. Alternatively, the gradual accumulation of sediments in the ice pits might homogenize the seabed, but this possibility cannot be assessed without knowing the ages of the ice pits. Further investigation is needed to understand how the characteristics of icebergs and those of the seafloor (e.g., sediment composition and thickness, bedrock topography) interact to shape ice pits and how they change through time.

The relative scarcity of algae outside of ice pits shows that the depression created by an ice pit clearly creates a depositional environment for negatively buoyant drifting algae. However, their dimensions can influence their algal contents, both quantitatively and qualitatively (i.e., biomass and species composition, respectively). Not surprisingly, estimates of total biomass increased with ice pit size, but the increase in area-specific biomass with ice pit size was unexpected. While this pattern could be due to thicker layers of algae accumulating in larger ice pits, our measurements of the thickness of the algal layer indicated little variation among ice pits of different sizes. Instead, the relationship between area-specific biomass and ice pit size appears to be related to the density of algal thalli within the algal layer. This layer, a loose mix of algal fragments, was largely composed of water, with algal tissue contributing up to 4% of its volume (i.e., ice pit “F”, Table 1). Factors that may have influenced the density of algal accumulations include the size of algal fragments (ranging in size from centimeters to meters; I. Garrido, unpublished data) or their morphology (e.g., branched vs. blades).

Ice pit dimensions also appeared to influence the species composition of accumulated drift algae. Diversity, at least as measured by species richness, differed between ice pits, with larger ice pits in deeper water having more species. This pattern is not surprising as larger pits contained more algal fragments, and there might be less loss of fragments from deeper locations. Species composition also varied among different sizes of ice pits, with different algal species characterizing small, medium, and large ice pits. For example, the three smallest ice pits had distinctly different species compositions than moderate and large ice pits. Despite being in the center of the study area, they lacked some of the species that dominated larger ice pits, including those nearby; instead, they were dominated by two rarely collected red algal species (*Ascoseira mirabilis* and *Georgiella confluens*) and one large brown algal species that do not grow in the immediate area (*Desmarestia anceps*). The underlying mechanisms producing this pattern are unclear but potentially might be related to the characteristics of the different algal species in terms of rates of deposition (e.g., buoyancy) or degradation in ice pits of different dimensions. Smaller ice pits might represent a recent deposition of more ephemeral species, whereas larger pits with larger, more common species may represent more stable assemblages. These findings further bolster the need for additional investigations of the interactions among algal source populations, fragment properties, and the transport of drift algae in this region.

Our two experimental manipulations shed light on certain dynamics of algal accumulation in ice pits. First, the high annual retention of PVC markers, our proxies for algal fragments, suggests that once drift algae are deposited in ice pits, further movement is minimal. Presumably, herbivory and microbial decomposition are then the primary processes of biomass loss. Second, when we experimentally removed all the algal biomass from ice pits, they remained empty over the short term in summer, but drift algal accumulations returned to pre-clearing levels within a year. Indeed, even though total biomass varied among the cleared ice pits, the annual deposition for each was consistent, suggesting that idiosyncratic features of individual ice pits determine the extent of the algal accumulation. Given our limited access to the site, it is difficult to determine when precisely drift algal deposition occurs, but the lack of accumulation in summer is not surprising as the primary disturbance processes that dislodge algae (i.e., waves, ice scour) are less likely to occur then. More likely is the fall, when periods of strong winds and storms would generate waves and currents that could increase hydrodynamic forces that can dislodge algae that have grown during the productive summer season. Spring is another possibility as large quantities of loose sea ice are formed at that time, thus increasing the likelihood of scouring the shallow rocky substrata where algae grow. Regardless, given the substantial annual deposition rates and the lack of advective loss, this system resembles a chemostat with an algal load capacity, which, once reached, remains dynamically stable; excess algae are removed by consumption and decomposition, and these losses are then compensated by seasonal input.

The precise source of drift algae remains unknown. At our site, drift algae in ice pits were composed mainly of red and brown algae, and most species found in ice pits were observed attached to nearby rocky substrata. The most abundant species found in ice pits (*Plocamium cartilagineum*, *Curdiaea racovitzae*, *Himantothallus grandifolius*) were also growing abundantly near the study area, suggesting a local source and limited transport. In support of this idea, other algae, such as *Desmarestia anceps*, which forms prominent algal forests elsewhere in Fildes Bay, were not well represented in the ice pits. Likewise, the composition of algae in ice pits is not necessarily a snapshot of local algal assemblages. Case in point, the rhodophyte *Palmaria decipiens* grew in only moderate abundance near the study site, and it was nearly absent in the 2018 data presented in this work (<4 g/m^2^). However, it was a dominant species in ice pits in the following year (I. Garrido and L. Johnson, personal observation) with no readily apparent explanation. These observations emphasize the complex nature of drift algal dynamics.

The degradation of algal detritus is assumed to be extremely slow in the soft bottom environments of Antarctica [35,70] compared to temperate waters (e.g., [66,86]) and is especially slow during the winter (June to September) [34,35]. Factors contributing to this slow degradation include low temperatures, limited mechanical damage due to low wave energy, extreme shade adaptation, low respiration rates, and secondary metabolites, which serve to protect the algae from grazers and bacterial development [87]. Moreover, measurements of the photosynthetic capacity of drift algae indicate that they can retain functional photosystems at those conditions, at least on the top of accumulations, whereas plants from the bottom of accumulations can show signs of reduced photosynthetic capacity [70]. Overall, these factors that slow the degradation of drift algae may contribute to the stability of this habitat over time, and their persistence may serve to dampen the seasonality in food supply by providing a more consistent source of food to the benthic fauna. In this sense, algal detritus can clearly play an important role in the food webs of Antarctic coastal waters, and the abundant fauna associated with this habitat (I. Garrido, unpublished data) clearly suggests that it may play an important role in retaining local primary production in shallow water and providing trophic links to the coastal food web. However, the time scale for its incorporation into the food web may be long, and very little is known about the utilization of this resource by higher trophic levels [88].

Accumulations of drifting algae in ice pits are expected to have different effects on abiotic and biotic conditions within them, depending on the temporal and spatial extent of algal depositions and local hydrodynamic conditions [63,67,89]. In the case of abiotic conditions, microbial degradation of drift algae can cause hypoxic or even anoxic conditions in the system [63]. In our study, oxygen sensors deployed inside and outside of the algal layer in an ice pit revealed stable conditions outside but lower and more variable ones inside, reaching a situation of hypoxia (I. Garrido, unpublished data). Conditions at the sediment surface were likely anoxic under the dense accumulations of drift algae observed here, although elsewhere in the Antarctic, thinner algal accumulations (approximately 4 cm) did not induce anaerobic conditions [70].

## 5. Conclusions

This work is a detailed glimpse into a poorly known nearshore habitat and provides a baseline for future observations. It also highlights the important role of ice scouring in Antarctic soft-bottom environments by both dislodging algae and creating ice pits into which much of that drift algae accumulates. Variations in the physical characteristics of ice pits influenced the quantity and species composition of algal accumulations in them, and their biomass appears to be dynamically stable through time. These features provide a unique but underappreciated ecological subsidy for soft-bottom habitats that may be important for structuring the benthic communities.

Ice scouring is a key disturbance process in polar regions that will be dramatically altered by climate change. Growing evidence indicates that accelerated warming in the Antarctic will lead to greater iceberg calving and increased production of drift algae. However, relevant information, such as calving rates of glaciers and the size distribution, longevity, and movements of icebergs, is still limited, and further research is required to understand how these factors will combine to affect the benthic assemblages of Antarctic marine ecosystems.

## Figures and Tables

**Figure 1 biology-12-00128-f001:**
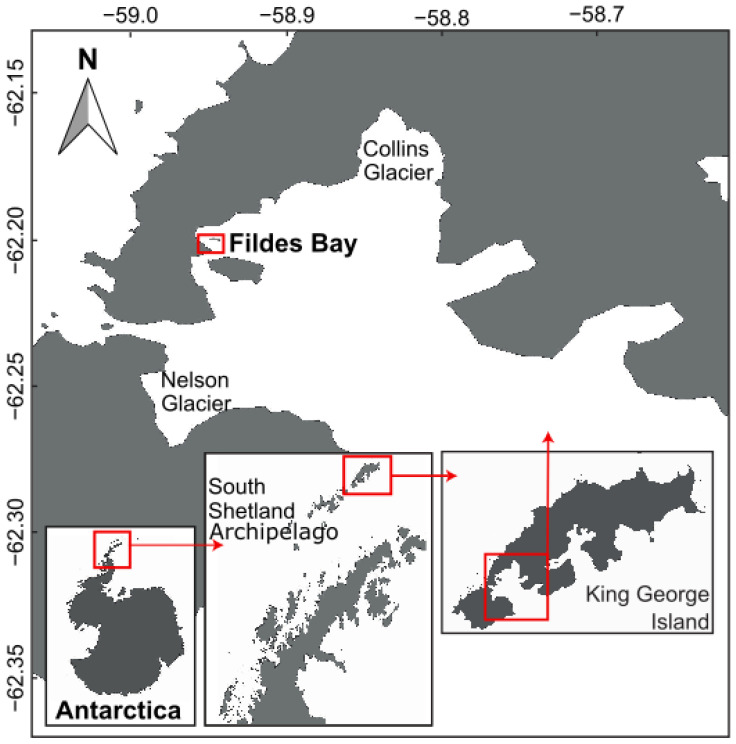
Location of the study area (approximately 62.20° S, 58.95° W) in Fildes Bay on the coast of King George Island in the northern region of the South Shetland Archipelago.

**Figure 2 biology-12-00128-f002:**
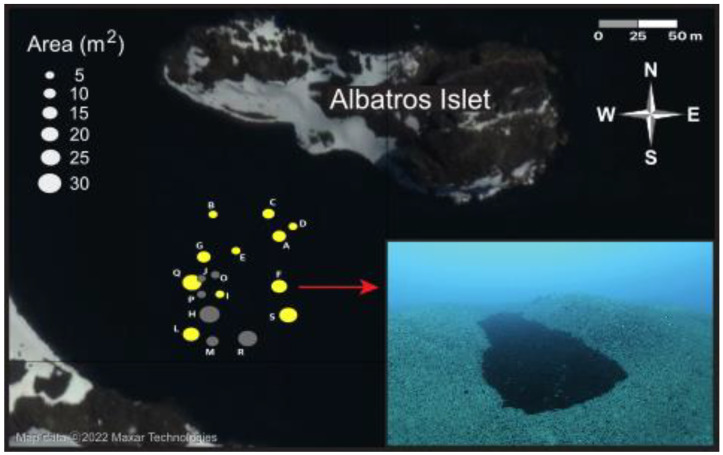
Distribution and size of the 17 ice pits documented in 2018. Algal biomass and composition were sampled with quadrats in eleven ice pits (yellow shapes). Ice pits were distinguished from the surrounding soft sediment seabed by distinct algal deposits within depressions (inset photo).

**Figure 3 biology-12-00128-f003:**
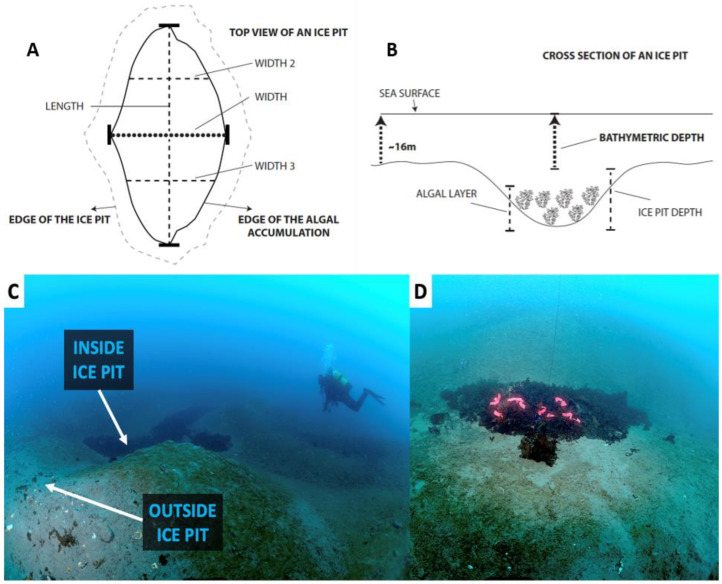
(**A**) Top view and (**B**) cross-section schematic representations of the dimensions recorded for ice pits. (**C**) Locations indicating inside and outside an ice pit (photo at 17 m from 2018). (**D**) Orange-colored straight-line polyvinyl chloride (PVC) flagging tape was deployed in ice pits to experiment with advective loss (photo at 15 m from 2018).

**Figure 4 biology-12-00128-f004:**
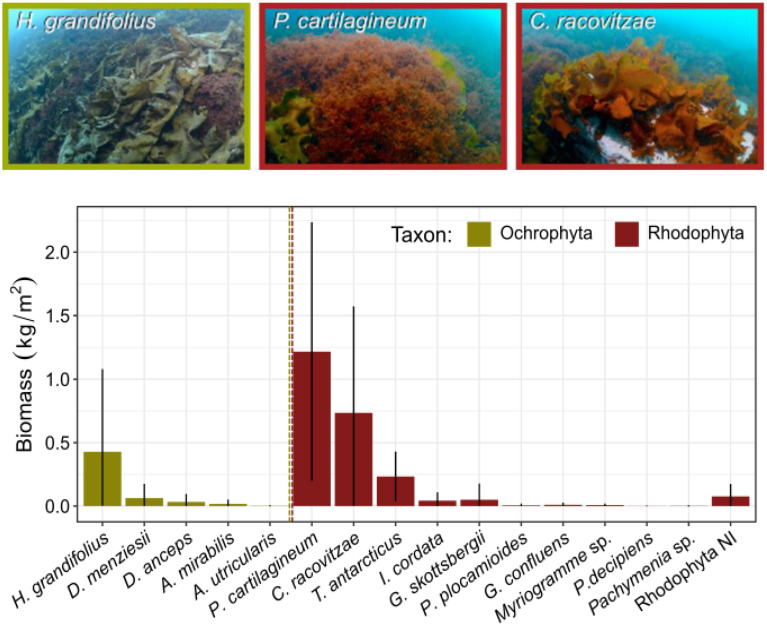
Area-specific biomass (kg/m^2^) of the 16 algal species observed inside ice pits based on quadrat samples collected from 11 ice pits in 2018. Full species names are provided in Table 3. Species are colored according to higher taxa; error bars represent the standard deviation around mean values. Rhodophyta NI is small, unidentifiable pieces of rhodophyte algae. Photo descriptions: *H. grandifolius*: a bed of *Himantothallus grandifolius* in Fildes Bay at a depth of 15 m. *P. cartilagineum*: an individual clump of the branched red algae *Plocamium cartilagineum* near Albatros Islet at a depth of 6 m. *C. racovitzae*: the red alga *Curdiaea racovitzae* at a depth of 7 m at Albatros islet.

**Figure 5 biology-12-00128-f005:**
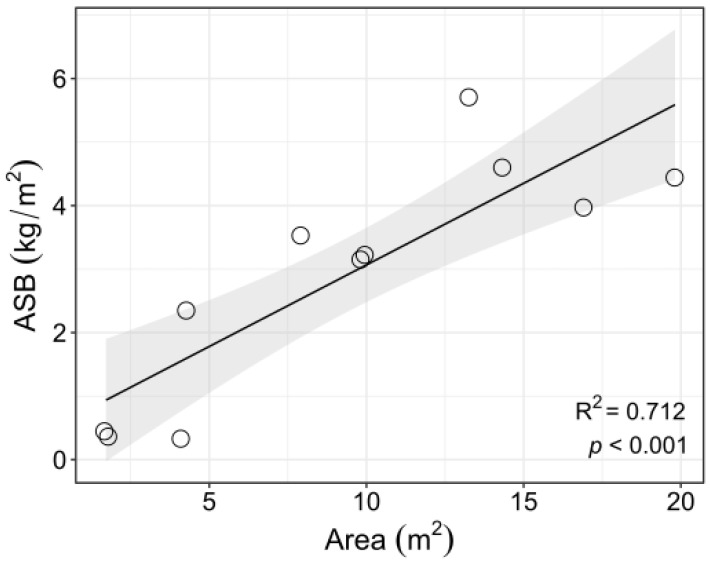
Area-specific biomass of drift algae in ice pits as a function of the ice pit area. Circle-like symbols representing the area-specific biomass of each sampled ice pit. The shaded ribbon spans the 95% confidence interval.

**Figure 6 biology-12-00128-f006:**
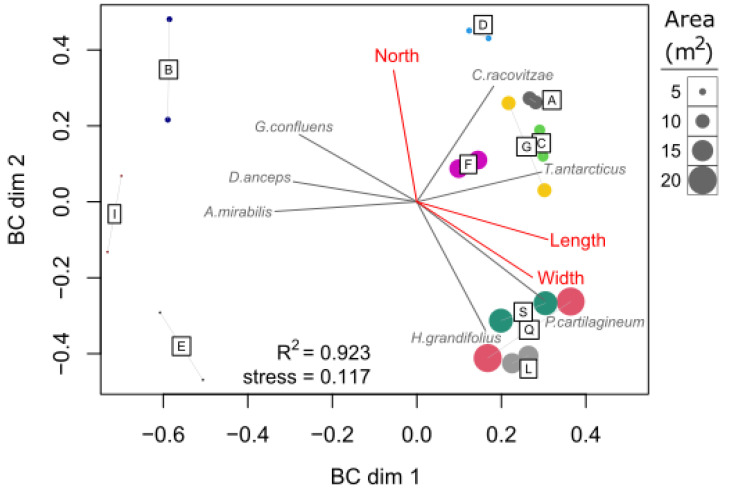
Non-metric multidimensional scaling (nMDS) ordination of the relative biomass of 16 algal species using Bray-Curtis dissimilarity. Each quadrat sample (two for each ice pit) is labeled and colored according to the ice pit and sized relative to the area (m^2^) of the ice pit. Vectors indicate algal species (gray) and environmental variables (red) with significant fit to the ordination at the *p* < 0.01 level.

**Figure 7 biology-12-00128-f007:**
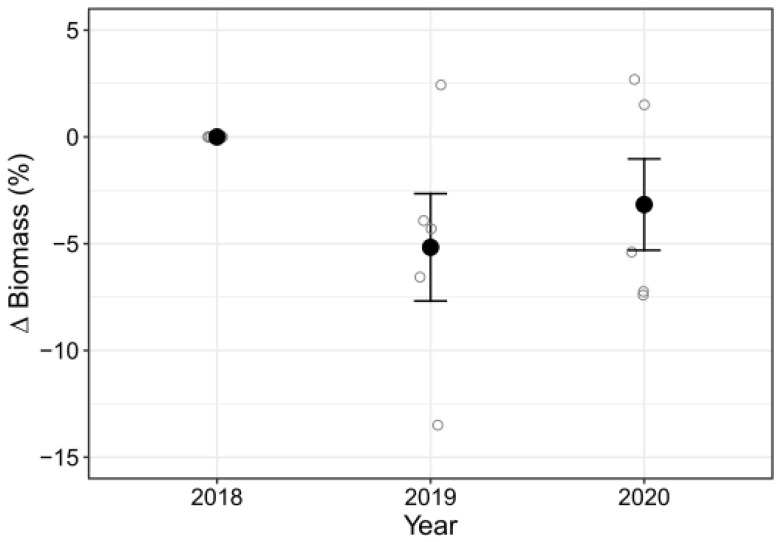
Percent change in the total algal biomass over two years, each time sampled in early January, one year after the manual removal of algae. Initial, pre-cleared 2018 biomass values ranged from 8.2 to 13.1 kg. Error bars span the 95% confidence interval around the mean (black circles) of the five experimental ice pits (open circles).

**Table 1 biology-12-00128-t001:** Geographical positions, dimensions, and algal biomass of ice pits. Six ice pits were not sampled for algal composition. For large ice pits, multiple length and width dimensions were measured, and the average values are shown here. Algal thickness was measured in the center of each ice pit; pit volume was calculated using a hemi-ellipsoid formula; and the Area-Specific Biomass (ASB) was based on the mean of two 0.25-m^2^ quadrats in each ice pit.

Icepit	Length (m)	Width (m)	Algal Thickness (m)	Algal Pit Area (m^2^)	Algal Pit Volume (m^3^)	ASB (kg/m^2^)	Algal Density (kg/m^3^)	Estimated Total Biomass (kg)
A	6.50	1.93	0.10	9.85	0.66	3.22	32.2	21.2
B	3.50	1.50	0.15	4.12	0.41	0.39	2.57	1.06
C	6.00	1.67	0.15	7.87	0.79	3.48	23.2	18.3
D	3.20	1.70	0.15	4.27	0.43	2.35	15.7	6.69
E	2.00	1.10	0.15	1.73	0.17	0.40	2.68	0.46
F	10.10	1.67	0.15	13.25	1.32	5.65	37.6	49.9
G	6.90	1.83	0.15	9.92	0.99	3.19	21.3	21.1
I	2.30	0.95	0.15	1.72	0.17	0.39	2.60	0.45
L	7.60	2.40	0.20	14.33	1.91	4.62	23.1	44.2
Q	8.70	2.90	0.15	19.82	1.98	4.44	29.6	58.7
S	6.80	3.17	0.15	16.93	1.69	4.01	26.7	45.2
H	6.50	4.27	0.20	21.80	2.91	-	-	-
J	2.70	1.50	0.15	3.18	0.32	-	-	-
M	3.60	2.80	0.15	7.92	0.79	-	-	-
O	1.00	1.00	0.15	0.79	0.08	-	-	-
P	1.30	0.80	0.15	0.82	0.08	-	-	-
R	7.20	3.40	0.20	19.23	2.56	-	-	-

**Table 2 biology-12-00128-t002:** Two Mantel tests using Euclidean distance matrices that test for correlations between pairwise similarity among ice pit shapes and their physical proximities (i.e., spatial autocorrelation) and between pairwise similarity among ice pit shapes and the species composition of sixteen algal species. Significance values are based on 999 permutations.

Eucl. Dist. X	Units	Eucl. Dist. Y	Units	Mantel r	*p* (perm)
Ice pit shape	m	Ice pit location	m	0.072	0.177
*pit length*		*N coordinate*			
*pit width*		*E coordinate*			
*pit depth*		*bathymetric depth*			
Ice pit shape	m	Algal composition	g/m^2^	0.571	*** 0.001**
*pit length*		*biomass sp. 1*			
*pit width*		*biomass sp. 2*			
*pit depth*		*…*			
		*biomass sp. 16*			

* Significant relationships at the *p* < 0.05 level.

**Table 3 biology-12-00128-t003:** Sixteen algal species were identified in this study, and their mean area-specific biomass (ASB; kg/m^2^) among 11 ice pits (two quadrats per ice pit).

Phylum, Species	ASB (kg/m^2^)	St. Dev (kg/m^2^)
**Ochrophyta**		
*Himantothallus grandifolius*	0.43	0.65
*Desmarestia anceps*	0.03	0.06
*Desmarestia menziesii*	0.06	0.11
*Ascoseira mirabilis*	0.02	0.04
*Adenocystis utricularis*	0.01	0.01
**Rhodophyta**		
*Plocamium cartilagineum*	1.22	1.02
*Trematocarpus antarcticus*	0.23	0.20
*Curdiea racovitzae*	0.74	0.84
*Iridaea cordata*	0.04	0.07
*Georgiella confluens*	0.01	0.02
*Gigartina skottsbergii*	0.05	0.13
*Pantoneura plocamioides*	0.01	0.02
*Palmaria decipiens*	0.01	0.01
*Pachymenia* sp.	0.01	0.02
*Myriogramme* sp.	0.01	0.02
**Chlorophyta ***		
*Monostroma hariotii*	<0.01	0.01

* Chlorophytes were found in ice pits, but due to their low biomass, they were not included in analyses.

**Table 4 biology-12-00128-t004:** Multiple regression analysis to predict if ice pit dimensions and relative position were good predictors of biological metrics. ASB: Area Specific Biomass.

Response	Predictor	Coefficient	St. Err	*t*	Pr (>|*t*|)	
Total Biomass (kg)	Coord. E (m)	−9.53	2.12	−4.51	0.011	*
	Coord. N (m)	−0.53	0.12	−4.28	0.013	*
	Pit Depth (m)	171.92	82.50	2.08	0.106	
	Bathym. Depth (m)	0.81	10.22	0.08	0.941	
	Coord. E: Bathym. Depth	0.59	0.13	4.54	0.011	*
	Pit Depth: Bathym. Depth	−10.60	4.81	−2.21	0.092	
ASB (kg/m^2^)	Coord. E (m)	0.03	0.01	4.70	0.018	*
	Coord. N (m)	−0.01	0.01	−2.27	0.108	
	Bathym. Depth (m)	0.77	0.44	1.74	0.181	
	Pit Depth (m)	7.08	2.94	2.41	0.095	
	Length (m)	0.48	0.06	7.99	0.004	*
	Width (m)	0.42	0.25	1.66	0.195	
	Bathym. Depth: Pit Depth	−0.42	0.17	−2.46	0.091	
Richness (S)	Coord. E (m)	0.99	0.42	2.34	0.079	
	Coord. N (m)	0.07	0.03	2.35	0.078	
	Bathym. Depth (m)	3.07	0.99	3.11	0.036	*
	Pit Depth (m)	2.03	0.99	2.05	0.110	
	Length (m)	1.12	0.34	3.34	0.029	*
	Coord. E: Bathym. Depth	−0.06	0.03	−2.34	0.080	

* Significant relationships at the *p* < 0.05 level.

## Data Availability

The data presented in this study are available upon request from the corresponding authors. The data are not publicly available because of restrictions related to privacy rights.

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
