# Peer review of "Drift Algal Accumulation in Ice Scour Pits Provides an Underestimated Ecological Subsidy in a Novel Antarctic Soft-Sediment Habitat"

_biology, 2023, doi:10.3390/biology12010128_

Round 1
Reviewer 1 Report
Brief summary
This article provide an interesting study on the formation of ice pits which lead to accumulation of marine algae in the ice pits. The community structure of algae in the ice pits was assessed by means of biomass, composition and diversity. In addition, the temporal dynamic of the algae accumulation and advective loss were also measured. There are many studies on the physical impacts of ice scouring on seabed ecosystem. However, this article discusses on different perspective and factor, in particular the formation of ice pits resulting from ice scouring, which subsequently creating new microhabitat govern by algae. This is rather new perspective to look at to and would make a good contribution in understanding the positive impacts of ice scouring on the benthic organisms. This article reads well with easy-to-understand sentences. However, in contrast to the Aims and Scopes of this journal suggest, this article was written rather to brief, especially in the Method and Result sections.
General concept comments:
1. The title with highlighting ‘benthos’ is misleading. The present study did not have any measurement on benthos / benthic organisms (commonly refers to the invertebrates). Therefore, the suggested new title is “Drift algal accumulation in ice scour pits provides an underestimated ecological subsidy in Antarctic soft-sediment habitat.”
2. Suggest to refer to several other more recent studies on drifting algae. E.g. https://link.springer.com/article/10.1007/s00227-021-04004-9
3. Generally, some parts of the methods are unclear. In particular of how the pits were measured (refer to subsection 2.1 and 2.2). Did the divers survey the transects are in every year (2018, 2019 and 2020)? Table 1 shows the information of the ice pits dimension, but it does not tell when the measurements were taken.
4. Description on the ice pits measurement should also cite any previous related work if any. If there is none, the authors should justify the methods were used. For instance, in measuring the ice pits, authors, the distinctive edge of algal accumulation was used instead of the actual physical dimension of ice pit. The authors are even acknowledging the underestimation but did not provide any justification to support the shortcomings.
5. The method section should also consistence in terms of writing style. Some sentences are passive structure while some are written in active sentence with personal pronouns.
6. Authors should write in details on how the algal identifications were carried out from sample collection, processing then the identification.
7. The methods for determination of advective loss should also be improved. Authors used the fluorescent fabric as proxy for drift algae. The measurements of the fabrics were given, but there was no explanation provided as to why such measurements were used. This section also does not describe how the fabrics were placed on the algae. The statistical analysis related to this loss was not described in subsection 2.5 (page 8).
8. The aims and scopes of this journal is for the authors to describe the work in details so that it can be reproduced by others. Therefore I would suggest the authors to pay a great attention to the Method section and thoroughly describe the methods used, including the experimental design.
9. Description of Figures and tables should be detailed e.g. state clearly what year(s) the figures show (figure 3), and what samples each dot represents in each year (figure 6).
10. In general, results section has a lot room of improvement. Somehow results are not described according to the methods. For instance, in temporal dynamic of the algal drift, measurements were taken 2 to 4 weeks after removal and in the following years. The results section failed to report and compare the findings in detail.
11. Authors use a lot of parentheses, where these can be replaced by a brief description when comparing results. For example in section 4 Discussion, the parenthesis in line 9 to 11 contains too much of info that merit the sentence to be written in 2-3 sentences.
12. Discussion section was articulated and easy to understand. The explanations correspond to the results section. However, this section lacks citation to support the arguments.
Specific comments
Abstract
The abstract is way too long (440 words) compared to the maximum length of about 200 words as in the guideline.
Intro
i. Page 2 Para 1 - Introduction to the disturbance associated with ice and glacier in Antarctica should cite a few more recent works.
ii. Page 2 Para 3 – Rephrase the part describing ice pits. In the 4th sentence of this para, ‘ice pits’ is resulting from the ice scouring process, whereas, in the 5th sentence, it sounds as if ‘ice scouring’ and ‘ice pits’ are two different things.
iii. Page 2 Para 4 - Suggest to add more works on the effects of ice scouring on benthos. (may check if relevant:
§ https://www.nature.com/articles/s41598-021-96269-9, https://www.ncbi.nlm.nih.gov/pmc/articles/PMC5226713/
§ https://research-repository.uwa.edu.au/en/publications/benthic-community-response-to-iceberg-scouring-at-an-intensely-di
§ https://www.cambridge.org/core/journals/polar-record/article/abs/understanding-the-link-between-sea-ice-ice-scour-and-antarctic-benthic-biodiversitythe-need-for-crossstation-and-international-collaboration/47E6DEA6CB112C7FF32CD26FFC6C73FC
iv. Page 2 Para 5 line 1 – Suggest to change to “ ...transport is driven mainly by currents ..”
v. Page 3 Para 1 line 7 – suggest to change “…even reproduction are possible”
vi. Page 3 para 1 line 7 to 9 – Rephrase the sentence beginning “Degradation is more likely, ..”. This sentence can be split into two sentences.
vii. Page 3 para 2 line 5 – suggest to change to “…, which subsequently provides food resource.
viii. Page 3 para 2 line 8 to 10 - please rephrase the last sentence.
ix. Page 3 para 2 – Please provide several citations for this para especially that discuss on the effects of algae in ice pits on the coastal ecosystem.
x. Page 3 para 2 – This paragraph needs a rewriting in order to describe the justification of the present work. Writers could also be more specific on what are the benefits of the present study rather than just mentioning about the potential ecological role and to provide baseline information for future changes. This para also lacks of relevant citation i.e. on ecological aspects. The only citation is a study by Siegert et lll, and yet this is on the general climatic issues.
Materials and Methods
i. Page 3, Study site – The first sentence should be rewrite. It is too long and difficult to understand. The use of parentheses, is also confusing.
ii. Page 3 and, Study site – Please provide information on the increased depth of the channel.
iii. Page 4 line 9 to 10 – Rephrase the sentence starting “Less abundant …”
iv. Page 5 Figure 1 – Suggest changing the presentation of the figure. Figs A, B and C can be combined to clearly show that the Figs C and B are zoomed images of A. Figs. D and E should be in separate figure as they do not describe ‘Study site’ as the subsection suggests.
v. Page 5 Section 2.2 – Suggest rephrasing the first sentence describing the transects. The use of ‘parallel’ word is rather confusing as those transects were perpendicular to the shoreline. It is also suggested that this description is supported by a figure.
vi. Page 5 Section 2.2 line 6 to 8 – The use of parenthesis is not suitable and making if difficult to understand the sentence. This part can be written in two sentences.
vii. Page 6 Figure 2 – This figure should also be separated with A&B and C&D for better describing the figure.
viii. Page 7 subsection 2.3 – If there were only 11 ice pits used for this study, then it is suggested table 1.1 also present 11 ice pits. There is no strong reason as to why 6 of the other icepits were not sampled
ix. Page 7 subsection 2.4 – The mechanism of how the fluorescent tape acted as drift algae is not clearly described. In addition, this specific technique would be much better to have cited previous work.
x. Page 9 subsection 2.5 last para – Please state what test was used to determine the algae lost.
xi. Page 9 subsection 2.5 last para – Suggest that the authors provide methods how ANOVA test was used, i.e. using pre-test to determine if the data are parametric or non-parametric before analysing the data using ANOVA.
Results
i. Page 8 subsection 3.1 – Suggest to indicate in the figure the divisions of the ice pits’ grouping as described (10 in the west, 5 in the east). The description in the parenthesis should also be rephrased.
ii. Page 8 subsection 3.1 – Add description in the text the correlation between pits dimension and algal composition correspond to the data presented in Table 2 (Mantel test)
iii. Page 9 subsection 3.2 para 1 – Suggest to change the second sentence to “……in the samples (Table 3), but there were only species, Plocamium carti-lagineum, Curdiaea racovitzae (both are rhodophytes), and Himantothallus grandifolius (a large, endemic phaeophyte) observed inside the ice pits, with overall contribution of 81.5%”.
iv. Page 10 – Please provide description of the 3 photos.
v. Page 11 figure 3. State whether the figure shows overall data of 3 years, or any specific year.
vi. Page 12 & 13 subsection 3.3 – The first 5 lines need rephrasing. In the beginning of the sentence, authors mention that the algae were restored. Yet the statistical test suggest otherwise.
Discussions
i. Page 15 para 4 – The explanation as to how larger icebergs created highly variable of length and width, but correlated ice pits is not clear. Perhaps authors can compare with the dimensions of ice pits formed by smaller ice?
ii. Page 15 para 5 line 5 to 9 – In “which could be due to either thicker or denser” argument, the ‘thicker’ factor seems to be unjustified here as the next argument only supports the ‘denser’ factor.
iii. Page 16 para 3 line 8 – Please explain what is the meaning of ‘earlier levels’ in algal accumulation.
iv. Page 16 para 3 line 12 to 15 – Please reassess the sentence. Shouldn’t the primary disturbance is more likely to occur in the summer, at least as compared to the winter? Perhaps the authors can add some references to support the arguments (extending to line 21) in this paragraph.
v. Page 16 para 3 line 5 to 7 – The deployment of oxygen sensors is not described in Method section, likewise, data related to this sensor are not presented. Yet, this is discussed in Discussion
vi. Page 17 – The last paragraph of Discussion is more suitable as Conclusion. This para is partly redundant with actual Conclusion in the next section.
Conclusion
Please provide general results in the Conclusion.
Bibliography
Reference format should be consistent throughout. There are several references’ titles written in ALL CAPS.
All the best. Thank you
Reviewer 2 Report
This study is descriptive and shows nicely how algal material accumulates in ice pit formations on the sea floor in Antarctica. These ice pits and the algal material in them seem to be quite persistent over time, and efforts to study the temporal development of the algal material give nice additional information. The degrading algal material likely serves as food and shelter for a number of organisms and increases the local biodiversity. However, I miss information on the invertebrate data, which obviously has been collected (referred to unpubl data), to be able to assess the actual importance of these habitats as potential biodiversity hot spots.
Some details to improve:
Start of materials and methods; some sentences need revision.
Nicely described area and materials and methods, fig 2 is informative.
In results, the following sentence is incomplete: “Their dimensions were, however, not strongly correlated with only length significantly correlated with width (r = 0.59, p = 0.01), and neither length (r = 0.14, p = 0.6) nor width (r = 0.36, p = 0.16) correlated with depth.” Tab 2 is unclear, is it all algal species tested against pit shape metrics or separate species? (biomass sp 1, biomass sp.2). Please clarify in tab text. The significance value in Tab 2 is not clearly mentioned and referred to in the results (comes much later under headline 3.2). To have the results in chronological order would improve readability.
In fig 3, what is NIredalgae and why is Monostroma not there?
In table 4, I am not convinced it is relevant to test only algal kg as it of course is dependent on pit size (I would skip as results are obvious). Kg/m2 sounds more relevant to test and discuss. And, add in table (and/or tab text) what ASB in table stand for (comes later in Fig 4 legend).
In fig 5, please, explain the coloring (yellow, blue etc.), I guess it is only to distinguish between different pits?
Fig 7 is redundant, the results are clear from the text.
I strongly suggest that data on invertebrates within the algae and in the sediment under the algae in the pits (vs controls) are added to this manuscript (sampling mentioned in materials and methods). Presently, this data is mentioned as unpublished so it obviously exists, but including it would show the potential significance of these algal accumulations in the ice pits and how the ecosystem may benefit/suffer from the accumulated algal material. Further, discuss how (if?) biodiversity/functioning of the system is affected based on results from invertebrate-studies.
Good to hear that O2 levels were controlled for, and that anoxia occurred was quite expected. How this affected the benthic community would naturally be nice to know and add to the potential significance for food webs/biodiversity of these algal pits. Likely the O2 concentrations in the algal sediment interface is also measured?
In the discussion, I was happy to read about the surrounding algal community and where the algae in the pits likely originated from.
Some repetition occur in the end of the discussion (climate change effects on future iceberg formation), and the entire discussion could to some extent be shortened (especially if invertebrate fauna is included).
